# Population structure and phylogenetic analysis of *Vibrio cholerae* non-O1/O139 by whole genome sequencing

Taylor Wells[1,2]*, Elizabeth González-Durán[3], Anthony M. Smith[4,5],
Swapan K. Banerjee[6], Sandeep Tamber[6], Natalie Knox[1,2], Celine Nadon[1,2]

**1** Public Health Agency of Canada, National Microbiology Laboratory, Winnipeg, Manitoba, Canada,
**2** Department of Medical Microbiology and Infectious Diseases, University of Manitoba, Winnipeg, Manitoba, Canada, **3** Instituto de Diagnóstico y Referencia Epidemiológicos "Dr. Manuel Martínez Báez" (InDRE), Secretaría de Salud, Mexico City, Mexico, **4** National Institute for Communicable Diseases, Division of the National Health Laboratory Service, Johannesburg, South Africa, **5** Department of Medical Microbiology, School of Medicine, Faculty of Health Sciences, University of Pretoria, Pretoria, South Africa, **6** Bureau of Microbial Hazards, Food and Nutrition Directorate, Health Canada, Ottawa, Ontario, Canada

* taylor.m.wells@phac-aspc.gc.ca

## Abstract

Toxigenic *Vibrio cholerae* serogroups O1 and O139 are well known for causing excessive diarrhea leading to devastating cholera epidemics and pandemics. Over 200 other serogroups, usually lacking the cholera toxin, are denoted non-O1/O139 *V. cholerae* (NOVC), and cause vibriosis leading to sporadic gastroenteritis and other extraintestinal infections. NOVC infections are not a notifiable disease in Canada and thus underreported. From 2010 to 2023, 160 cases and a small 2018 outbreak were reported in Canada caused by NOVC, provoking considerable public health concern. In this study, 242 Canadian *V. cholerae* isolates were sequenced, characterized and compared with over 1500 other *V. cholerae* isolates from around the world to determine their genetic relationships. All Canadian NOVC and two O139 isolates lacked the cholera toxin-producing genes typically harbored by pathogenic O1 and O139. All 14 Canadian O1 isolates were identified from travel-related cases as members of the toxigenic 7[th] pandemic lineage, whereas one O139 isolate was acquired domestically. Phylogenetic analysis based on core genome single nucleotide polymorphisms classified the Canadian isolates into five clades. Eight new lineages of NOVC, denoted CAD1–8, were identified from the Canadian isolates. A new lineage was defined as clusters formed by three or more isolates in the phylogeny. These lineages were comprised of isolates from clinical origin alone, environmental origin, or a mixture of both. Some lineages spanned multiple years and regions. CAD-2 was comprised of clinical and environmental isolates associated with the 2018 outbreak. Several virulence genes were detected among NOVC, including hemolysins, toxins and secretion system encoding genes. A proportion of virulence genes differed between isolation source (clinical or environmental) and clinical manifestations (gastrointestinal or

**Data availability statement:** All sequence assemblies are available from the NCBI GenBank database and BioSample accession numbers are listed in S1 Table. Newly sequenced genomes are available under NCBI's BioProject PRJNA1230206. Interactive figures and associated metadata can be found here: https://microreact.org/project/cLqNBRRphmN-t1aoBeh9jmJ-fig-5 and https://microreact.org/project/nGCx8mpZEkvX2jZW2QriJj-fig-2. All other relevant data are within the manuscript and its Supporting information files.

**Funding:** This work was funded in part by the Government of Canada Shared Priority Project, Genomics Research and Development Initiative (GRDI)-AMR and Health Canada A-base. Whole genome sequencing of South African isolates was made possible by support from the SEQAFRICA project, which is funded by the Department of Health and Social Care's Fleming Fund using UK aid. The views expressed in this publication are those of the authors and not necessarily those of the UK Department of Health and Social Care or its Management Agent, Mott MacDonald. The funders had no role in study design, data collection and analysis, decision to publish or preparation of the manuscript.

**Competing interests:** The authors have declared that no competing interests exist.

extraintestinal). Our study identified environmental sources of NOVC with the potential to cause human infection. Tracking the emergence of NOVC with pathogenic potential is essential for understanding the risk to Canadians.

## Introduction

*Vibrio cholerae* is a Gram-negative bacterial pathogen that naturally inhabits freshwater, estuarine and marine environments worldwide [1]. *V. cholerae* is well known for causing cholera, a food and waterborne disease responsible for roughly 2.9 million cases, and 95,000 deaths worldwide each year [2]. There are over 200 serogroups of *V. cholerae* characterized by the O-antigen component of the outer membrane lipopolysaccharide [3]. Members of serogroups O1 and O139 typically carry the key determinants of cholera pathogenicity, toxin coregulated pilus (TCP) and cholera toxin (CT), precursors for intestinal colonization and severe watery diarrhea, respectively [3]. *V. cholerae* O1 and O139 hardly ever occur in developed countries with adequate access to clean water and sanitation; sporadic cases and outbreaks are typically associated with travelers who have visited endemic areas [4,5].

Serogroups other than O1 and O139 are collectively referred to as non-O1/O139 *V. cholerae* (NOVC) and usually do not produce the cholera toxin and thus do not cause cholera. Instead, NOVC and other non-cholera *Vibrio* species such as *V. parahaemolyticus*, *V. vulnificus*, and *V. alginolyticus* cause a variety of bacterial infections in humans referred to as vibriosis. The clinical manifestations of vibriosis depends on the route of infection, host susceptibility and the type of *Vibrio* species [6]. Illness caused by NOVC often leads to mild or moderate gastroenteritis and can be transmitted through the consumption of raw or undercooked seafood [7], a common practice and delicacy in many parts of the world. In addition to gastrointestinal infections, vibriosis can cause a wide range of extraintestinal manifestations after water exposure, via a cut or scrape on the skin, or by entering through the ear, leading to infections of wounds [8], skin and soft tissue [9,10], ear [11,12], and blood (bacteremia, secondary septicemia) [13–15]. Sporadic cases of necrotizing fasciitis [16,17], meningitis [18], keratitis [19], urinary tract infections [20,21] and oral infections [22] have also been reported. Diagnosis of NOVC from extraintestinal infections can be challenging; infections might not be recognized or potentially underdiagnosed because of its uncommon description in the literature and limited physician knowledge.

NOVC usually causes sporadic infections and, in the past, has caused rare, small-scale outbreaks [3,23,24]. More recently, however, several countries have reported vibriosis outbreaks caused by NOVC, including Chile [25], China [26], and The United States [27]. In 2018, an outbreak of vibriosis caused by NOVC occurred on Vancouver Island, British Columbia in Canada [28]. The outbreak was associated with the consumption of herring eggs, a traditional food source for First Nations communities, that were harvested from the Vancouver Island coast [29]. Although the outbreak was small (three laboratory confirmed cases), the incident drew a lot of attention, as it was the first documented outbreak of NOVC in Canada, highlighting the emergence of this pathogen and the potential risk to Canadians.

Over the last two decades, PulseNet Canada has had great success using molecular and genomic epidemiology for foodborne disease surveillance, outbreak detection, and response [30–32]. Although, these efforts have been limited to the top foodborne bacterial pathogens, including *Listeria monocytogenes,* verotoxin-producing *Escherichia coli, Shigella,* and *Salmonella* species. On top of the national genomic surveillance conducted by PulseNet Canada, the National Enteric Surveillance Program (NESP) tracks confirmed cases of a wide variety of enteric bacteria, parasites and viruses reported by each province weekly [33]. However, the surveillance information collected from these additional enteric pathogens is limited. Over the last decade, there has only been an average of 11 laboratory-confirmed cases of NOVC in Canada each year [33], though the rising water temperatures due to climate change is conducive to the growth and survival of *Vibrio* species and is expected to contribute to the rise in vibriosis worldwide [34]. In Canada, the emergence of *V. cholerae* in estuaries and coastal regions indicates the possibility of producing contaminated seafood for human consumption [35]. Moreover, the health of First Nations is particularly impacted in communities that utilize traditional harvesting and food preparation practices of fish and seafood [29].

Until now, no molecular or genomics study of NOVC in Canada has been reported; this study will characterize the variation in genomic features that could be used for future surveillance, outbreak detection, and response. This study aims to characterize a collection of clinical and environmental NOVC collected in Canada and elsewhere to determine the virulence potential, antimicrobial resistance patterns, and population structure as well as uncover the phylogenetic relationships. These data on the occurrence, diversity, and genomic characteristics of NOVC causing illness in Canada and around the world will enable an up to date picture of the risks posed by this organism as well as preparedness using genomic tools.

## Materials and methods

### Sampling and dataset

To accurately describe the NOVC population in Canada, isolates representing the NOVC within Canada, outside Canada, and O1/O139 from both Canada and international locations were included. The primary dataset consisted of 226 NOVC isolates collected by PulseNet Canada Provincial Public Health Laboratories during passive surveillance and routine reference services in Canada during the years 1998–2023 (Fig 1). Isolates belonging to serogroup O1 (n = 14) and O139 (n = 2) were collected in Canada from 2017–2023 were also included in this study; these formed relevant comparators useful for understanding NOVC in Canada. An additional 38 NOVC genome sequences were shared by PulseNet International partners in South Africa and Mexico. Approximately 8,500 publicly available genome sequences downloaded from the National Center for Biotechnology Information (NCBI; up to December 2023) and presumed to belong to serogroups O1, O139 and NOVC were initially selected for this study. *V. cholerae* O1 strain N16961, the first sequenced genome of *V. cholerae,* was used as the reference throughout [36]. Public genomes were collected for various purposes (i.e., research, passive surveillance, routine reference services and outbreak detection) and do not represent a well-defined epidemiological cohort.

### Whole genome sequencing and quality analysis

Of the samples collected in Canada, the genomes of 88 were already sequenced via the Illumina MiSeq platform prior to the start of this study. The remaining isolates were streaked from fresh culture or frozen stock onto Columbia blood agar base with 5% sheep blood with or without enrichment in alkaline peptone water (pH = 8.2). Single colonies surrounded by typical green to clear zones were subcultured onto brain heart infusion agar in preparation for DNA extraction. Genomic DNA was extracted using the Qiagen DNeasy Blood and Tissue Kit and the concentrations were determined using the Qubit dsDNA broad range assay kit (Invitrogen). The genomic DNA was sent for paired-end whole genome sequencing (WGS) using the Illumina MiSeq Platform and 600 cycle V3 kits at the Genomics Core at the National Microbiology

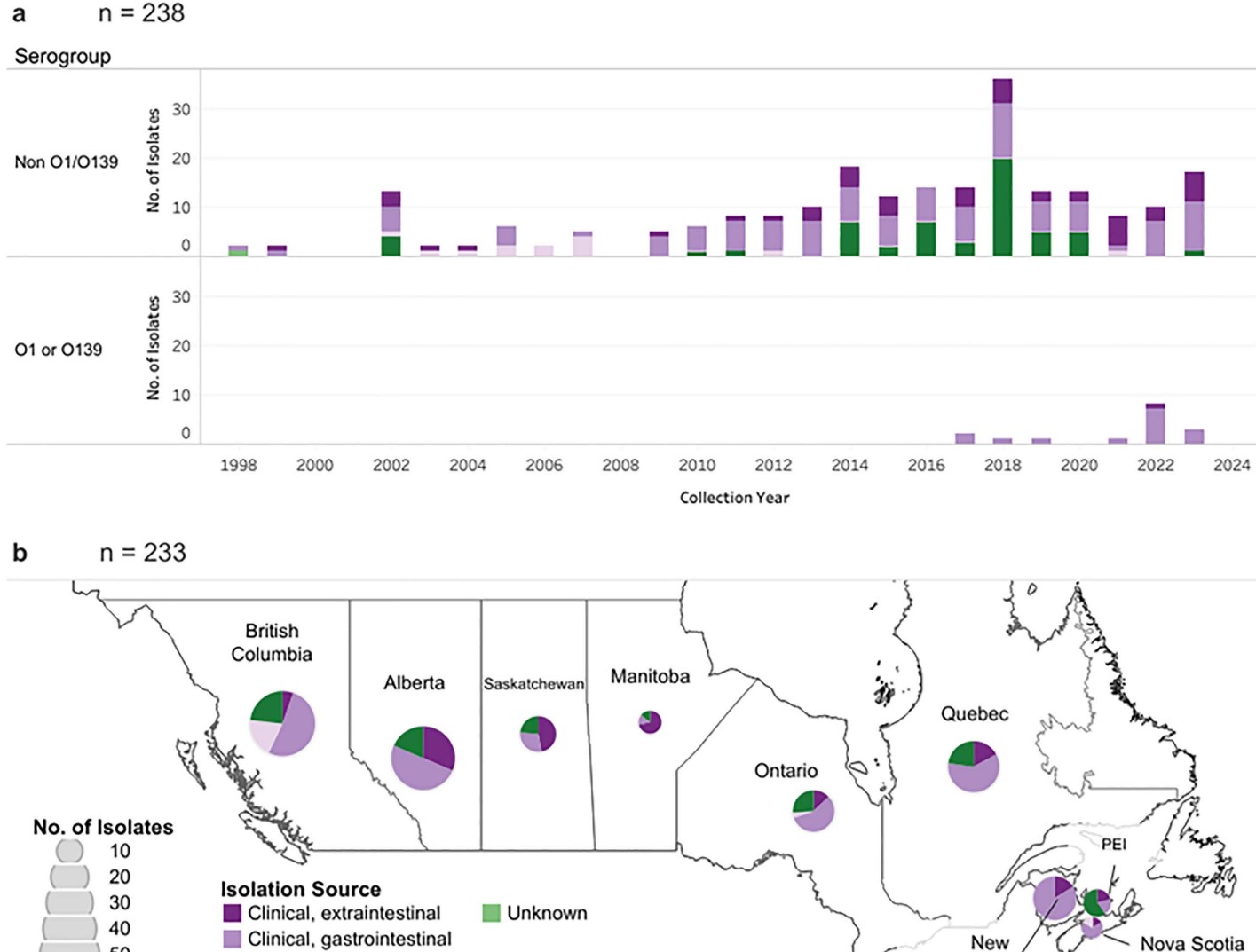

**Fig 1. Geographic and temporal origin of the 242 *V. cholerae* collected in Canada.** a. Collection year of *V. cholerae* isolates according to serogroup. Four NOVC isolates did not have a collection year recorded. b. *V. cholerae* isolates were received by provincial laboratories and sent to the NML in Winnipeg, Manitoba for characterization and whole genome sequencing. The diameter of each circle scales with the number of isolates to have been collected from the indicated province. Geographic data were unavailable for nine isolates. The bar graphs (a) and pie charts (b) are colored by the original source of isolation. It should be noted that the peak in NOVC isolates in 2018 are a result of the small outbreak on Vancouver Island, British Columbia which required increased environmental sampling to trace the outbreak source. Map was created using Tableau Public 2021.4 [37] using base map and data from OpenStreetMap and OpenStreetMap Foundation..

Laboratory (NML), Winnipeg, Canada. Raw sequence data was deposited into the Integrated Rapid Infectious Disease Analysis (IRIDA) platform (internally hosted instance) for downstream analysis [38].

The quality of the raw reads was assessed using FastQC v0.11.9 [39], FastP v0.20.0 [40] and MultiQC v1.9 [41], and assembled using Shovill v1.1.0 with SPAdes v3.14.1 [42]. Assembled genomes were evaluated using the Quality

Assessment Tool (QUAST; v5.0.2) using *V. cholerae* N16961 as the reference genome [43]. The completeness and contamination levels of the genome sequences were computed using CheckM v1.1.3 [44]. Species identity was determined using the Genome Taxonomy Database Toolkit (GTDB-Tk) [45]. Assembled genomes downloaded from NCBI were run through the mikrokondo pipeline v0.1.0 [46], an all-encompassing workflow that provides routine bioinformatics tasks (e.g., contamination and quality assessment of assemblies, taxonomic classification). All genomes were annotated using Prokka v1.14.6 [47].

Initially, 4,800 *V. cholerae* genomes of good quality were kept using the following criteria: coverage ≥ 40x, genome fraction > 95%, NrContigs ≤ 150, and N50 > 40,000. A low N50 (i.e., high fragmentation of assemblies) and low coverage results in poor allele calling [48]. All genomes were nearly complete (≥ 97%), with medium to low levels of contamination (≤ 7%) as determined by checkM. Publicly available genomes distantly related to NOVC were removed in addition to isolates with incomplete epidemiological data.

The final data set resulted in 1,791 genomes collected from 62 countries and spanning 87 years, from 1937 to 2023, with the majority collected after 2000. Altogether, there were 492 NOVC, 181 O1/O139 and 1,118 isolates that did not have serogroup information provided. The genomes of all analyzed isolates are available in GenBank under the accession numbers listed in S1 Table, with newly sequenced genomes available under the BioProject ID PRJNA1230206.

## Phylogenetic analysis

Parsnp v1.2 [49] was used to reconstruct the maximum likelihood phylogenetic tree based on core genome single nucleotide polymorphisms (cgSNPs) of the 242 genomic sequences of isolates collected in Canada and using *V. cholerae* N16961 as the reference genome. The -x flag was used to enable filtering of SNPs in recombinogenic regions as identified by PhiPack v1.1 [50]. Default settings were used for all other parameters. In order to build a population structure of *V. cholerae* specific to Canada, the cgSNP alignment of the 242 genomes was used as the input for performing Bayesian hierarchical clustering and partitioned using the Dirichlet Process Mixture model with fastBAPS (v1.2) [51]. The final clustering was indicated by a partition of the FastTree phylogeny [52], while the other parameters were set as default.

Genome assemblies were uploaded to Vibriowatch [53] implemented in the Pathogenwatch web tool [54]. Vibriowatch uses the *V. cholerae* multilocus sequence typing (MLST) scheme based on the seven housekeeping genes [55] and the core-genome MLST (cgMLST) scheme based on 2,443 core genes [48]. To place the Canadian genomes in the global context, cgMLST profiles of all 1,791 genomes were generated and the number of core-genome alleles distinguishing each *V. cholerae* isolate was calculated using cgmlst-dist [56].

## Comparative analysis

The presence/absence of virulence genes and genomic islands was assessed using the BLASTN algorithm [57] against the choleraFinder database [58] with key modifications; the database was supplemented to include prominent virulence genes described in the Vibriowatch database [54] as well as major exotoxins found in *Vibrio* species according to the Virulence Factor Database (VFDB) [59]. Genes were obtained from the tool's repositories; CholeraFinder [60], Vibriowatch [61] and VFDB [62]. A gene/segment was considered present if the overall hit coverage and percent identity were at least 60 and 95%, respectively. Genotypic AMR was inferred using Vibriowatch by querying assemblies against a library of antimicrobial resistance genes and variants using AMRsearch [63] with an identity threshold of 90% and coverage of 98%. Lastly, the presence and absence of plasmids were screened in Vibriowatch using the IncTyper tool [64].

## Data interpretation

Clustering thresholds were defined based on the cgMLST scheme established by Liang and colleagues [48] (Table 1). Any two isolates that showed a cgMLST difference of 0–3 alleles were considered genetically identical, whereas a difference of ≤ 40 alleles were considered very closely related. The sublineage threshold based on 133 allelic differences creates

**Table 1. Clustering thresholds of *V. cholerae* based on cgMLST alleles.**

| No. alleles | Relationship | Clustering |
|---|---|---|
| ≤ 3 | Identical | Identical |
| ≤ 40 | Very closely related | Outbreak |
| ≤ 133 | Closely related | Sublineage |
| ≤ 1000 | Related | Lineage |

clusters similar to traditional 7-gene MLST where isolates are considered closely related. Finally, isolates with < 1,000 allelic differences will be considered related and may belong to the same lineage. CAD lineages were defined as three or more related isolates in the Canadian phylogeny.

### Statistical analysis

Comparisons between two independent groups (e.g., clinical vs. environmental, extraintestinal vs. gastrointestinal NOVC infection, and serogroups) and gene presence or absence were undertaken using Pearson's chi-squared test (Fisher's exact test for groups of $N < 5$). Meanwhile, Kruskal-Wallis rank sum test was performed for identifying group level differences across virulence and AMR profiles. All analyzes were performed using R v4.2.1 [65] and p-values < 0.05 were considered statistically significant.

### Data visualization

Metadata were visualized and maps were created using Tableau Public 2021.4 [37]. Phylogenetic trees were visualized using Microreact [66] and in R v4.2.1 [65] using ggtree v3.4.4 [67]. Gene presence/absence matrices were visualized in R using pheatmap v1.0.12 [68]. Where figures were edited manually, this was performed using Paint 3D v6.2405.19017.0.

## Results and discussion

### The incidence of vibriosis caused by NOVC is rising in Canada

Overall, reported infections caused by *V. cholerae* are rare in Canada; with an average of approximately 10 cases of vibriosis (serogroup NOVC) and two of cholera (serogroups O1 or O139) documented annually (Fig 1a). This data shows an apparent upward trend in the number of NOVC isolates over the last two decades. Similar reports of increasing NOVC are found in the United States [69] and across Europe [70]; the rising sea surface temperature as a result of global warming is favorable to the growth and survival of *Vibrio* species and this is a likely cause of shifting patterns of vibriosis [71]. In Canada, it is unclear whether the rising incidence in vibriosis is caused by more NOVC illness in general or as a result of heightened clinical surveillance and improved diagnostics. Enhanced environmental surveillance of seafood from Canadian coastal waters [72] and retailers [73,74] has certainly contributed to the increase in NOVC isolates originating from environmental sources [35]. The peak of NOVC isolated in 2018 coincides with the vibriosis outbreak that occurred in British Columbia, which resulted in enhanced environmental sampling specifically targeting *V. cholerae* to trace the outbreak source.

### Clinical characteristics and geographical patterns of NOVC isolates

Of the total NOVC isolated in Canada, 75% (n = 169) were of human clinical origin with no reported links to international travel, 25% (n = 56) were environmental (i.e., animal, food or water sources), and only one isolate did not have a recorded origin (S2 Table). Water samples were collected from natural aquatic environments (e.g., lake, ocean), and food samples were collected from food sources harvested from the environment for human consumption (e.g., herring eggs, mollusks, shrimp). In this study, clinical samples refer to human biological specimens that were collected for diagnostic purposes.

While there was no clinical data available to provide detailed insight on the disease associated with the clinical isolates; reasonable inferences on the type of infection can be derived from the source of isolation. Stool samples are typically collected when trying to establish a diagnosis from gastroenteritis; inflammation of the stomach and intestines leading to symptoms like nausea, vomiting, abdominal cramps and diarrhea. Over half of the NOVC isolates that were of clinical origin were associated with gastrointestinal infections (inferred by the specimen type) (64%, n = 108), while one quarter were presumed to be associated with extraintestinal infections (28%, n = 48) including isolates derived from the ear (n = 17), blood (n = 13), tissue/wound (n = 11), urine (n = 6) and eye (n = 1). Thirteen (8%) NOVC isolates of clinical origin did not have an isolation source recorded. Gastroenteritis is the most common clinical presentation of NOVC infections worldwide [75,76]; however, heat-wave associated vibriosis has been linked to a rise in extraintestinal manifestations, due to the increase in recreational exposure to contaminated water [8,77]. As seen in this study, wound and ear-related infections are among the most common forms of extraintestinal vibriosis as they are more susceptible to penetrating water [70]. Most surprising here is that there were six NOVC isolates from urine; urinary tract infections are rare clinical manifestations of vibriosis [20,21]. This study also provides the third report of a *V. cholerae* isolated from an eye sample of a supposed eye infection; the first was endophthalmitis [78] and the second was keratitis [19]. This highlights the expanding spectrum of NOVC infection and suggests that more attention should be directed towards NOVC as a suspect in eye and urinary tract infections.

The *V. cholerae* isolates collected in this study were geographically distributed across nine out of the ten Canadian provinces (Fig 1b). Nearly half (46%) of the clinical infections caused by NOVC were concentrated along The West Coast (British Columbia) and The Atlantic Region (New Brunswick, Nova Scotia and Prince Edward Island). These results are unsurprising considering the long and highly populated coastlines [34], proximity to salt water sources, as well as the popularity of both commercial and recreational marine fishing and shellfish harvesting in these provinces. Interestingly, Alberta had the highest number of clinical NOVC infections (n = 44, 26%) and together with Saskatchewan and Manitoba, the Prairie Provinces had the highest proportion of extraintestinal infections. These three provinces are essentially landlocked, with the exception of Manitoba's North Eastern coast on the Hudson Bay, but this climate is sub-arctic and not conducive to *V. cholerae* survival or growth. Over the past 60 years, the national average temperature has risen by 2.8°C, with the highest increase observed in Northern Canada (4°C) [79]. Aside from generalized warming patterns, the rising sea levels and expansion of coastlines are contributing to the geographical expansion of *Vibrio* species [80] and will continue to play a major role in shaping the human population exposed and the increasing risk of infection [34].

## NOVC carry multiple virulence factors in Canada

All NOVC isolates in this study were non-toxigenic and lacked the genes encoding CT (*ctxAB*) and TCP (*tcpA*; Fig 2). The *ctxAB*, zonula occludens toxin (*zot*), accessory cholera toxin (*ace*) and repeat sequence transcriptional regulator (*rstR*) genes are prominent virulence factors that make up the CTXϕ bacteriophage [81]. One NOVC isolate carried *zot*, *ace*, and *rstR*, despite the absence of *ctxAB* genes or the CTXϕ receptor, TCP. Other investigations have reported toxin-producing NOVC [7,82] and in some cases have been implicated in small outbreaks [24]. For reasons unknown, toxin-producing NOVC has yet to spark the same epidemic response as O1 and O139. This may in part be attributed to the improved hygiene, access to clean water and adequate sanitation facilities, which all contribute to slowing the spread of disease where NOVC is most heavily reported [24]. The epidemiological success of toxigenic NOVC may also rely on how these strains acquire cholera toxin genes (i.e., horizontal transfer or O-antigen biosynthesis) and the genomic configurations of the donor isolates (i.e., 7PET/toxigenic or non-7PET/non-toxigenic) [24,83].

Virulence genes encoding hemolysin (*hlyA*), hemagglutinin protease (*hapA*), thermolabile hemolysin (*tlh*), agglutinin-like sequence (*als*), and repeats-in-toxin A toxin (*rtxA*) as well as regulatory elements including LuxS-mediated quorum sensing (*luxS*), alternative sigma factor (*rpoS*), and ToxR regulatory protein (*toxR*), were each present in over 90% of all *V. cholerae* isolates collected in Canada. Other virulence factors detected among NOVC included genes encoding

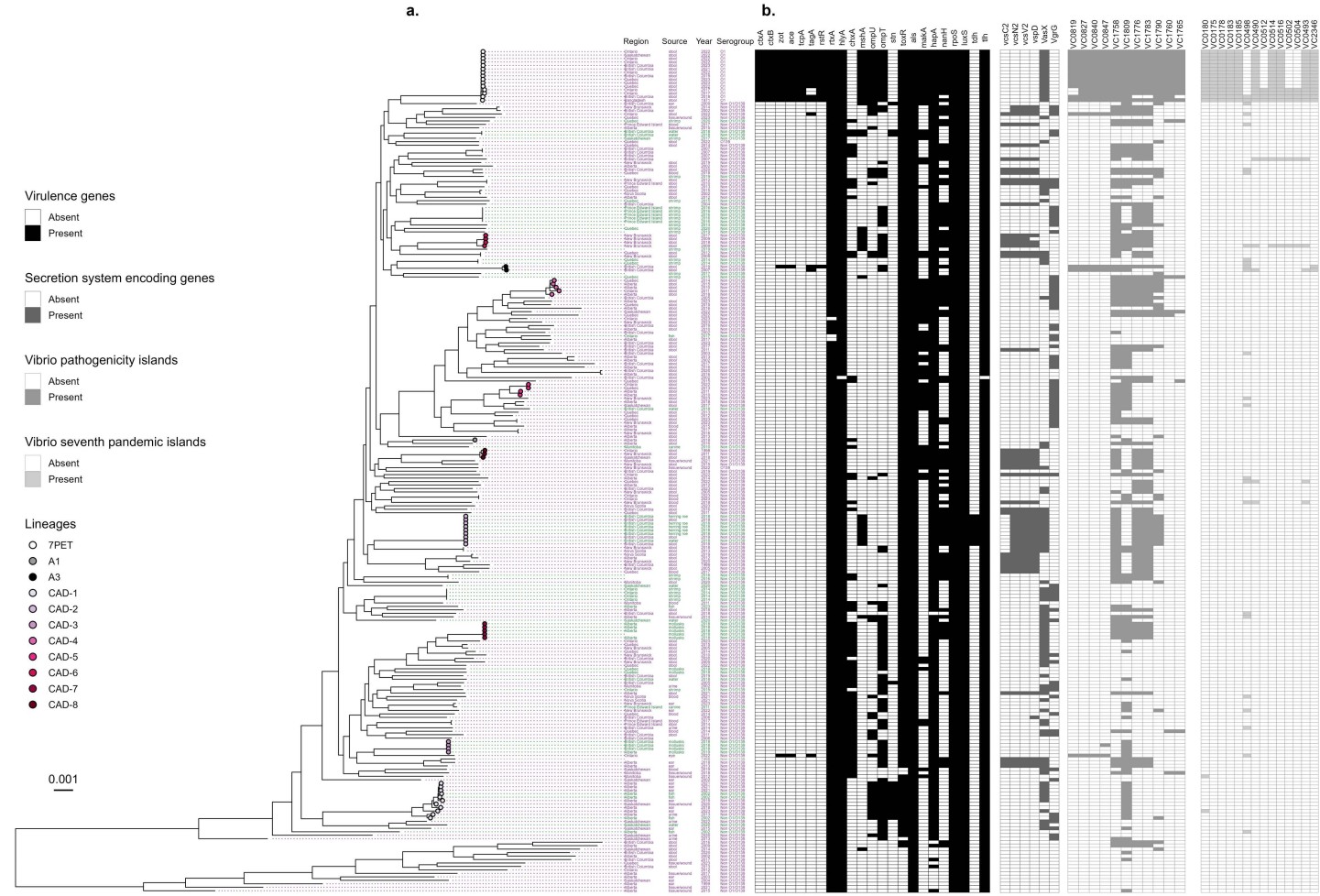

**Fig 2.** *In silico* cgSNP tree of Canadian *V. cholerae* isolates with virulence gene profiles a. **Maximum likelihood tree showing phylogenetic relationships between 242 clinical and environmental isolates with lineage, region within Canada, isolation source, collection year, sero-group, and b. Heatmap showing the presence (colored in box) and absence (white box) of virulence and secretion system encoding genes,** *Vibrio* **pathogenicity islands, and** *Vibrio* **seventh pandemic islands.** Tip labels describing isolation origin, source, year and serogroup were colored purple to represent isolates of clinical samples, green to represent environmental samples and black if sample type was not known. Aside from the reference (*V. cholerae* O1 strain N16961; Bangladesh), all samples were isolated in Canada. Tip labels were left blank when data was not provided. Scale bar denotes the number of mutations per variable site. An interactive version of this figure is available on Microreact (https://microreact.org/project/nGCx8mpZEkvX2jZW2QriJj-fig-2).

neuraminidase/sialidase (*nanH*; 39%), outer membrane protein (*ompT*; 37%), type six secretion system (T6SS; *vasX*, 35%; *vgrG*, 32%), cholix toxin (*chxA*; 31%), motility-associated killing factor A (*makA*; 25%), T3SS (*vspD, vcsN2, vcsV2,* 21%; *vcsC2,* 15%), *ompU* (15%), mannose-sensitive hemagglutinin pilus (*mshA*; 10%), heat-stable enterotoxin (*stn*; 7%), thermostable direct hemolysin (*tdh*; 4%) and mucinase (*tagA*; 2%).

Only four and two NOVC isolates contained a complete set of *Vibrio* pathogenicity island 1 (VPI-1) or VPI-2 genes, respectively. Sixty-nine percent (n = 157) of NOVC carried at least one gene encoded with VPI-1 or VPI-2. Only two NOVC isolates carried any gene belonging to the *Vibrio* seventh pandemic island 1 (VSP-1). In both cases, this gene was VC0180, which encodes a putative protein of unknown function. VC0498 is predicted to encode ribonuclease HI, and was the most frequently detected gene of VSP-2 in NOVC (10%). One NOVC isolate carried a nearly complete (88%) VSP-2

island. Most surprising here is that two NOVC isolates carried the 7th pandemic (7PET) gene sequence (VC2346) that is used to distinguish 7PET from classical strains of *V. cholerae*. This gene is a component of the genomic backbone that is nontransferable (i.e., is not part of a genomic island or plasmid), thus confirming that these two NOVC isolates are progenitors of a 7PET strain [84]. Certain virulence factors such as hemolysin, cholix toxin, repeats-in-toxin, secretion systems and pathogenicity islands are also frequently found in the O1 and O139 serogroups [85]. This data provides further evidence of the pathogenic potential of non-toxigenic NOVC and warrants prospective surveillance of this pathogen.

## Environmental and clinical NOVC have different virulence gene profiles

The diverse virulence profiles of NOVC observed in this study have also been reported in previous work [58,86]. In total 84 different virulence profiles were observed among all NOVC isolates from Canada (Fig 2). Furthermore, the clinical isolates could be distinguished from the environmental isolates based on the differences in virulence gene profiles (Fig 3a, $p < 0.05$). Previous reports have both agreed [58,87] and disagreed with these findings [82,88] based on the authors own domestic population of NOVC. In Canada, the environmental isolates that originated from food samples displayed the most similar virulence profile to those of clinical isolates (Fig 3b). Whereas the virulence profiles of clinical isolates differed the most from environmental isolates from water and animal sources.

On average, clinical and environmental NOVC isolates each possessed approximately 11 virulence-associated genes. NOVC isolated from the environment had a significantly higher proportion of *mshA* (21%, $p < 0.01$), *ompT* (52%, $p < 0.01$), *toxR* (100%, $p < 0.05$), *hapA* (100%, $p < 0.05$), and *tdh* (11%, $p < 0.01$) compared to clinical isolates (6%, 31%, 91%, 93%, 2%, respectively). All isolates with the exception of one from clinical origin bore the hemolysin gene (*hlyA*). Similarly, most possessed *rtxA* with the exception of two clinical and one environmental isolate. A total of 16 NOVC isolates of clinical origin were missing the virulence regulator gene, *toxR*, which may contribute to diminished pathogenicity in these isolates. In contrast, there was a significantly higher proportion of *makA* (30%, $p < 0.05$) and T3SS encoding genes (*vcsN*, 24%, $p < 0.05$; *vcsV*, 24%, $p < 0.05$; *vcsC*, 20%, $p < 0.001$; *vspD*, 24%, $p < 0.05$) among clinical isolates compared to environmental isolates (11%, 11%, 0%, 11%, respectively). Similar to TCP, the *mshA* gene encodes a type IV pili that plays a role in attachment and biofilm formation of *V. cholerae* on biotic and abiotic surfaces, increasing their infectivity and survival [89]. The *hapA* gene encodes activity that simultaneously aids in the detachment of *V. cholerae* and penetration into the intestinal mucosal barrier during initial infection [85,90].

The presence and distribution of virulence factors in NOVC originating from clinical cases may be associated with their ability to cause disease. Isolates from probable extraintestinal infections had the highest proportion of outer membrane proteins (*ompU*, 40%, $p < 0.0001$; *ompT*, 52%, $p < 0.01$), and the heat-stable enterotoxin encoding gene (*stn*, 19%, $p < 0.0001$) when compared to gastrointestinal infections (7%, 24%, 0%, respectively). The role of outer membrane proteins in *V. cholerae* pathogenesis is not well understood; however, their critical role in bile resistance, environmental signaling, virulence factor expression and intestinal colonization have been suggested [91]. Awasthi and colleagues [92] suggested that the cholix toxin may be associated with extraintestinal infections because of the extensive damage to internal organs observed in mouse models. Although this study observed a higher proportion of *chxA* in isolates from probable extraintestinal infections (33%, $p > 0.05$) compared to those from clinical isolates (23%), the difference was not statistically significant. Additionally, cholix toxin plays a role in environmental fitness and survival [93] and has mostly been found in environmental reservoirs [94]; aligning with the transmission route for extraintestinal infections of NOVC [95]. However, the present study did not observe a statistical difference in the proportion of *chxA* in environmental isolates (36%, $p > 0.05$) compared to clinical isolates (28%). The heat-stable enterotoxin (NAG-ST) is a major enteric toxin of *V. cholerae* that is closely related to the heat-stable toxin produced by enterotoxigenic *E. coli* [96]. Although most studies report a low prevalence NAG-ST, which is encoded by the *stn* gene [58,82,88,97], others have detected *stn* in approximately one quarter of NOVC isolated from aquatic environments (i.e., seawater, estuary, sediment, plankton and oysters) [7,98] indicating foodborne and waterborne exposure are both possible exposure routes of *stn* positive NOVC. In this study, *stn* was not frequently detected among food samples (Fig 3b).

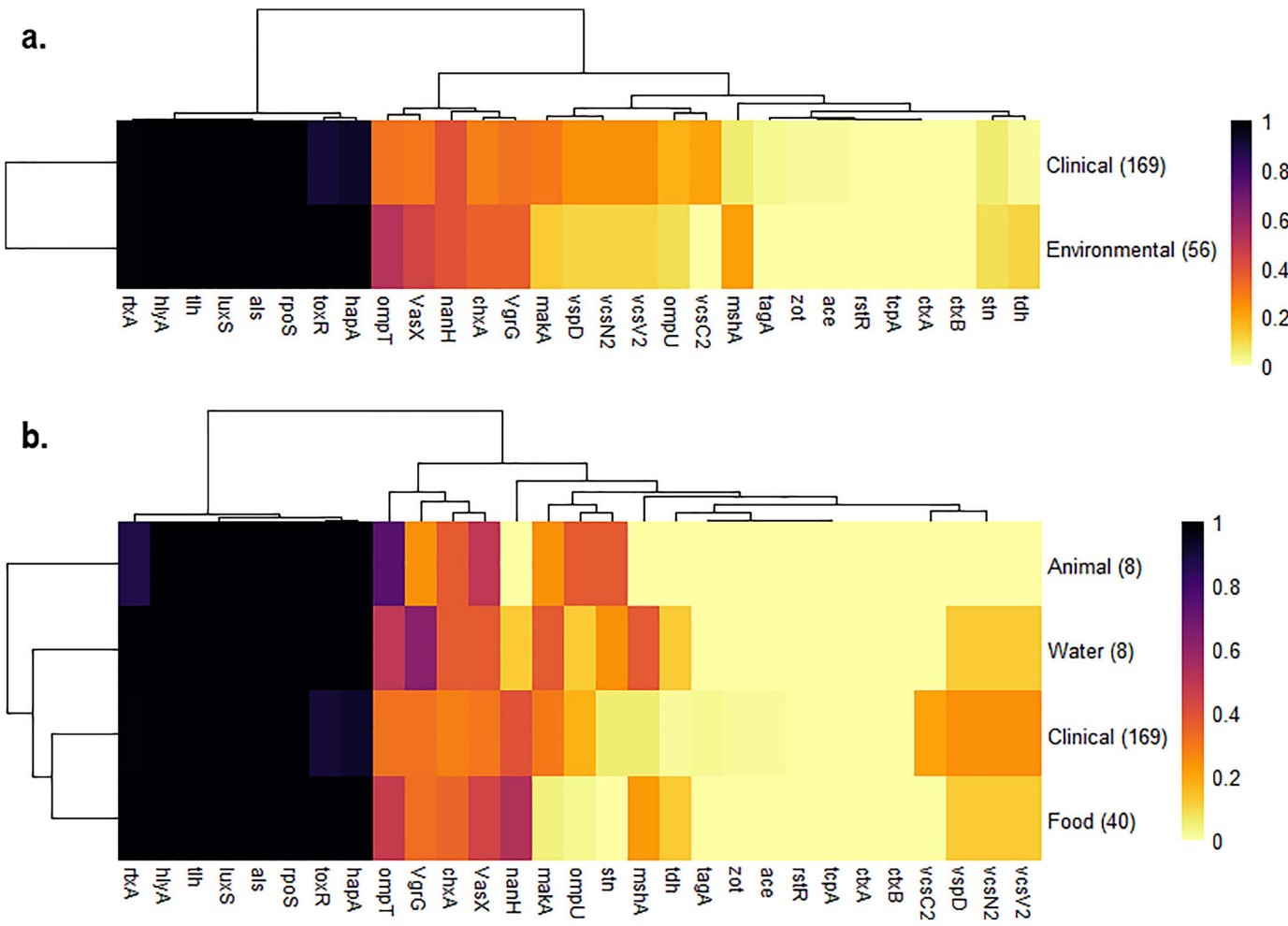

**Fig 3. Proportion of virulence genes detected among clinical and environmental NOVC in Canada.** a. NOVC by host information. b. NOVC by detailed host information. (*) represents genes with statistically significant differences $p<0.05$ between groups. One NOVC isolate did not have a recorded origin.

In contrast, a higher proportion of *makA* (39%, $p<0.001$), *nanH* (47%, $p<0.01$) and *toxR* (95%, $p<0.01$) genes were identified in NOVC from gastrointestinal infections than from extraintestinal infections (10%, 19%, and 79%, respectively). The motility-associated killing factor encoded by the *makA* gene is a newly discovered cytotoxin in *V. cholerae* with proven toxicity to zebrafish [99] and cultured human colon carcinoma cells [100]. Neuraminidase encoded by the *nanH* gene secretes sialic acids which provides the sole source of nutrition for *V. cholerae* in the intestine [101], which may aid in bacterial survival and colonization [94]. These results suggest that both *makA* and *nanH* in *V. cholerae* may favor the gastrointestinal route of infection; more studies should be completed to determine their role in *V. cholerae* pathogenesis of humans.

Secretion systems are protein structures used by many Gram-negative bacteria to deliver toxins and effector proteins in host cells. Genes encoding the T3SS are relatively common in NOVC and have previously been associated with diarrhea [102,103]. In this study T3SS encoding genes were found mainly in NOVC isolated from clinical cases, especially those isolated from probable gastrointestinal infections (i.e., isolated from stool samples; 23–29%). To a lesser extent T3SS genes were also detected in NOVC isolated from probable extraintestinal infections (15–17%) and environmental sources

(0–11%) which is probably attributed to their role in environmental adaptation [103]. T3SS are not commonly found in serogroup O1 and O139 isolates, whereas T6SS are more frequent [86,87,104]. In this study, the T3SS was not found in any of the serogroup O1 isolates but was detected in one serogroup O139 isolate collected from a wound infection. All of the serogroup O1 and O139 isolates and 26% of the NOVC isolates only carried the *vasX* gene which encodes the effector protein of the T6SS; the structural component VgrG that injects effectors into the host cell was missing [105]. The presence of VgrG may not be necessary for the entry and toxic effects of VasX on the host cells [106]. Both *vasX* and *vgrG* genes were found in twenty isolates from both clinical (n = 14) and environmental (n = 6) sources. Only three NOVC isolates from probable gastrointestinal infections harbored complete sets of both T3SS and T6SS encoding genes, of which two only differed by two alleles despite having been isolated from different years and provinces.

Four clinical isolates of NOVC harbored a complete VPI-1, only one of which was collected from a probable extraintestinal infection. The *tagA* gene encodes a mucinase and is frequently found in toxigenic serogroup O1 isolates as it is encoded within the VPI-1 and is positively co-regulated with CT and TCP [107]. In this study, *tagA* was found in four NOVC clinical isolates, which likely resulted in one extraintestinal and three gastrointestinal infections. Two of the clinical isolates that harbored *tagA* were isolated in the same province within the same three-month period but differed by over 2,200 alleles. In addition, *zot* and *ace* were detected in two *tagA*+ NOVC isolates, one from a probable extraintestinal infection and one from a gastrointestinal infection, from different years and provinces and with over 2,200 allelic differences in the core genome.

VPI-1 encoding genes were mainly identified in NOVC from clinical isolates from all different types of isolation sources. Only one NOVC isolate originating from an environmental sample harbored a single VPI-1 gene. All genes that encode VPI-2 were identified in NOVC isolates from both clinical and environmental origins. Only one VSP-1 gene was found in two NOVC isolates, both of which happened to be isolated from probable extraintestinal infections, but were otherwise not closely related according to cgMLST analysis (> 2,000 allelic differences). Genes encoding VSP-2 were found mostly in clinical isolates; only the VC0498 gene of VSP-2 was identified in NOVC originating from the environment (n = 3) and was also the most frequently found VSP-2 gene in NOVC isolated from probable extraintestinal (n = 9) and gastrointestinal (n = 19) infections.

## Several lineages of NOVC are circulating in Canada

Five major clades were identified using Fastbaps, which was consistent with the topology of the tree shown in Fig 4, with an interactive phylogenetic tree available at https://microreact.org/project/nGCx8mpZEkvX2jZW2QriJj-fig-2. This study confirmed the high genetic diversity that is typically observed among NOVC [108] and all of these isolates were dispersed among clades one, two, four and five. Within the five major clades, the phylogeny also revealed 10 distinct subclades of NOVC, referred to as lineages. Collectively, three NOVC isolates belonged to the A1 and A3 lineages recently named for contributing to epidemic and endemic disease in Argentina between 1993–2004 [109]. Similar to the findings of Dorman et al. [109], the Canadian isolates belonging to A1 and A3 were of clinical origin and belonged to serogroup NOVC. Both isolates belonging to A3 contained partial segments of CTXφ, VPI-2 and VSP-2, a complete VPI-1 and the 7PET marker gene (VC2346) and only differed by 230 alleles. The A3 isolates from this study did not harbor the T3SS, which is inconsistent with the A3 isolates from Argentina. Although *V. cholerae* tends to have a slow mutation rate [110], these isolates were collected over an average of 15 years after the original Argentinian isolates and thus may have lost the T3SS encoding genes and their associated virulence; although these isolates were still capable of causing gastroenteritis.

A total of 44 isolates (18%) were identified as belonging to eight new lineages of NOVC and were denoted CAD1−8 (Fig 4), where CAD stands for 'Canada'. A new lineage was defined as subclades formed by three or more isolates in the phylogeny. The CAD lineages identified in this study contained isolates that were of clinical origin alone (CAD-4,5,6,8), environmental origin (CAD-3,7), or both (CAD-1,2) and in some cases occurred in different years (CAD-1,3,4,5,6,8) and regions (CAD-1,3,4,5,8), suggesting that these represent populations of NOVC currently unique to the Canadian dataset (Fig 2).

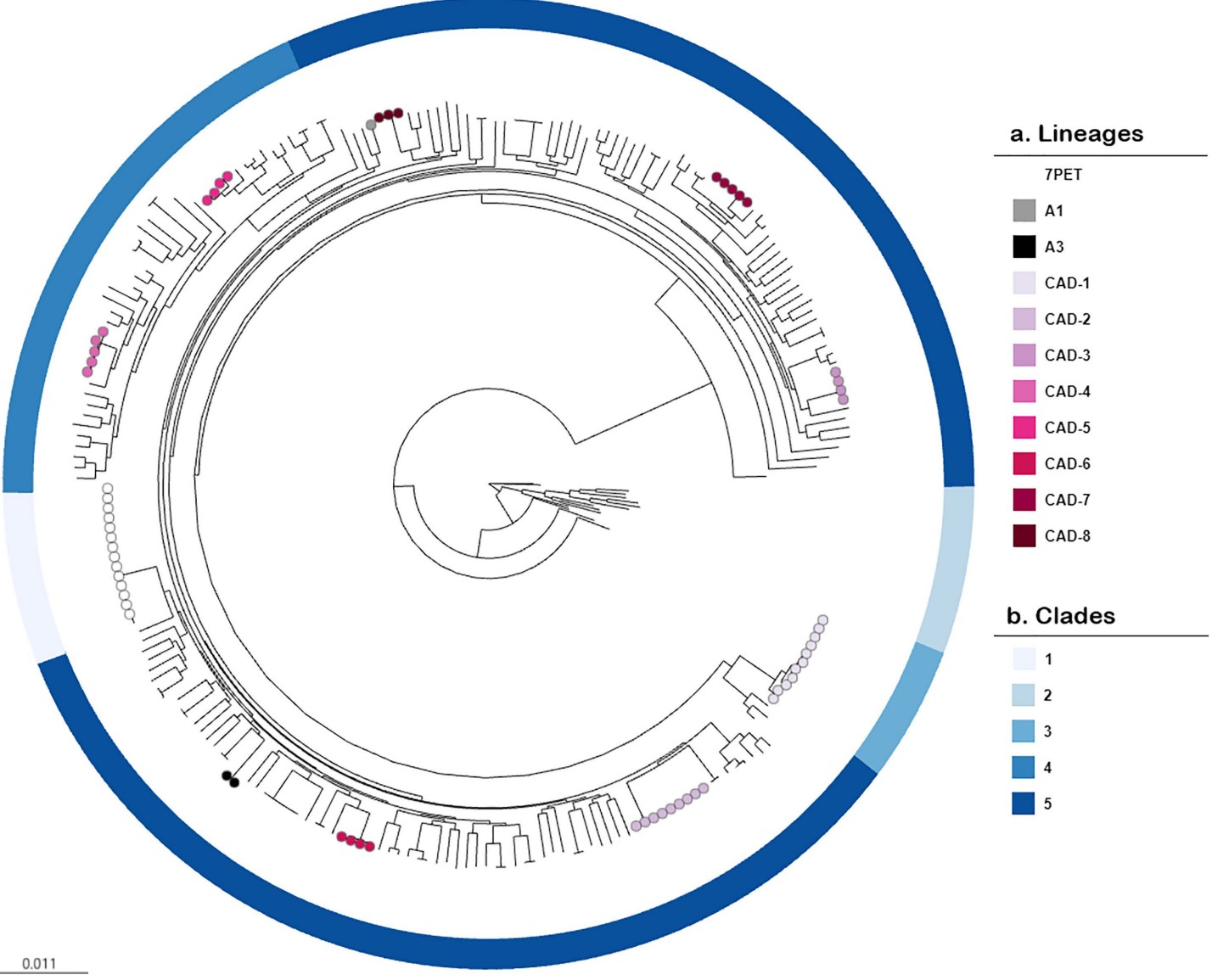

**Fig 4. Maximum likelihood tree of 242 Canadian _V. cholerae_ isolates based on non-recombined cgSNPs. a.** Tree nodes and colors represent lineages. The absence of a node means the isolate was not assigned to a lineage. **b.** The ring represents the five major clades as determined by Fast-Baps. An interactive phylogenetic tree is available on Microreact (https://microreact.org/project/nGCx8mpZEkvX2jZW2QriJj-fig-2).

### The CAD-2 lineage caused a point source outbreak in Canada

CAD-2 was associated with the only documented outbreak of NOVC in Canada which occurred in March 2018 and was restricted to two coastal areas on Vancouver Island (an island lying off southwestern mainland British Columbia) [29]. The outbreak investigation collected three clinical isolates and 10 environmental isolates from the two areas. Phylogenetic analysis revealed that five of the environmental isolates from herring eggs and one from a marine water sample, were identical to the human isolates (0–2 allelic differences), consistent with a point source outbreak. Domman and colleagues [111] characterized this disease pattern when isolates cause a short term, sporadic outbreak often associated with

foodborne infection and localized to coastal regions. The four remaining environmental isolates collected during the outbreak investigation originated from marine water samples collected in close proximity to the outbreak source but differed by more than 2,000 alleles across the core genome compared to the outbreak isolates. These results highlight the high diversity of *V. cholerae* sampled within a small geographical area.

The virulence profile of all nine CAD-2 isolates were identical. Isolates belonging to this lineage were non-toxigenic (lacking *ctxAB* and *tcpA*), but carried additional accessory virulence genes, including *hlyA*, *mshA*, *rtxA*, *toxR*, *als*, *hapA*, *nanH*, *rpoS*, *luxS*, *tdh*, and *tlh*. Interestingly, CAD-2 isolates were the only in this study to carry the *tdh* gene. Thermostable direct hemolysin (TDH), encoded by the *tdh* gene has been linked to hemolytic activity, cytotoxicity, cardiotoxicity and enterotoxicity [112] and is a major virulence factor of pathogenic *V. parahaemolyticus* [113,114]. To a lesser extent, *tdh* has also been reported in *Vibrio* species such as *V. mimicus*, NOVC, *V. hillisae*, *V. diabolicus*, and *V. alginolyticus* [115,116]. Fever and bloody stool with mucous is more common in patients with *V. parahaemolyticus*, but these symptoms are considered rare among NOVC infections [117]. It has been proposed previously that TDH may also be responsible for this rare clinical presentation in NOVC infections [116], although there is no direct evidence supporting this possibility. Future studies should focus on NOVC isolates carrying the *tdh* gene to investigate the pathogenic potential of TDH in NOVC infections.

CAD-2 was the only lineage to carry genes encoding both T3SS and T6SS. The T3SS and T6SS have been shown to be responsible for intestinal colonization and pathogenicity, respectively, suggesting a significant role in virulence during human infections [25]. Several NOVC outbreaks causing gastroenteritis in Chile [25], China [26], Nigeria [118] and the United States [27] have been associated with T3SS and/or T6SS and thus could be considered as molecular risk markers and may be useful in future surveillance activities. None of the CAD-2 isolates harbored any VSP islands or VPI-1 genes, whereas a partial VPI-2 region was found.

None of the outbreak isolates were closely related to any publicly available *V. cholerae* genomes. One isolate collected in 2006 in Sweden from a wound infection caused by serogroup O150 differed by 418 alleles and therefore could belong to the same lineage as CAD-2. The Swedish isolate was assigned to sequence type (ST) 774, while the CAD-2 isolates are a novel ST, differing from ST 774 only by the *purM* gene. Unlike CAD-2, this isolate did not carry *tdh*, but several other virulence gene similarities were found, including *hylA, mshA, toxR, nanH, hapA,* T3SS and T6SS. Several wound infections- including three fatalities- caused by NOVC were documented during the warm summer months in Sweden in 2006 [119]. These results suggest CAD-2 isolates collected from an outbreak of seafood-borne NOVC gastroenteritis may also have the potential to cause severe wound infections, if exposed to seawater.

## NOVC can be an ongoing cause of disease over long periods of time

The isolation source of the clinical isolates belonging to CAD-4, 5, 6 and 8 were all from stool samples, and likely caused gastroenteritis. Unlike the short, point-source outbreak observed by CAD-2, these lineages tend to be associated with cases of disease spread over very long periods of time and across geographic areas [111]. Similar disease patterns have been described among the Gulf Coast and MX-3 lineages of NOVC [120]. CAD-4, 5 and 8 caused sporadic cases uniformly distributed over several years and provinces, and were the only NOVC lineages to carry the *makA* gene (Fig 2). CAD-4, 5, and 8 lineages also harbored a combination of at least two additional virulence factors including *nanH*, and genes encoding a T3SS or T6SS. Further, one clinical isolate from CAD-4 only differed by 79 alleles to another clinical isolate collected from the Dominican Republic; and thus may also be a member of this lineage (Fig 5a).

The four clinical isolates that made up CAD-6 were also collected over several years (2009–2018), but unlike CAD-4, 5, and 8, were all collected from the same province (New Brunswick). CAD-6 isolates harbored several virulence genes including *mshA*, *ompT*, *nanH*, and T3SS encoding genes. Despite the difference in isolation years, three of the four isolates in this lineage were very closely related (4–10 allelic differences), indicating a recurring or ongoing contamination source in the area. In addition, CAD-6 isolates were only 15–23 alleles from an environmental isolate collected from a water sample in Germany. While clinical isolates are capable of spreading across the world, it is unusual for a very closely

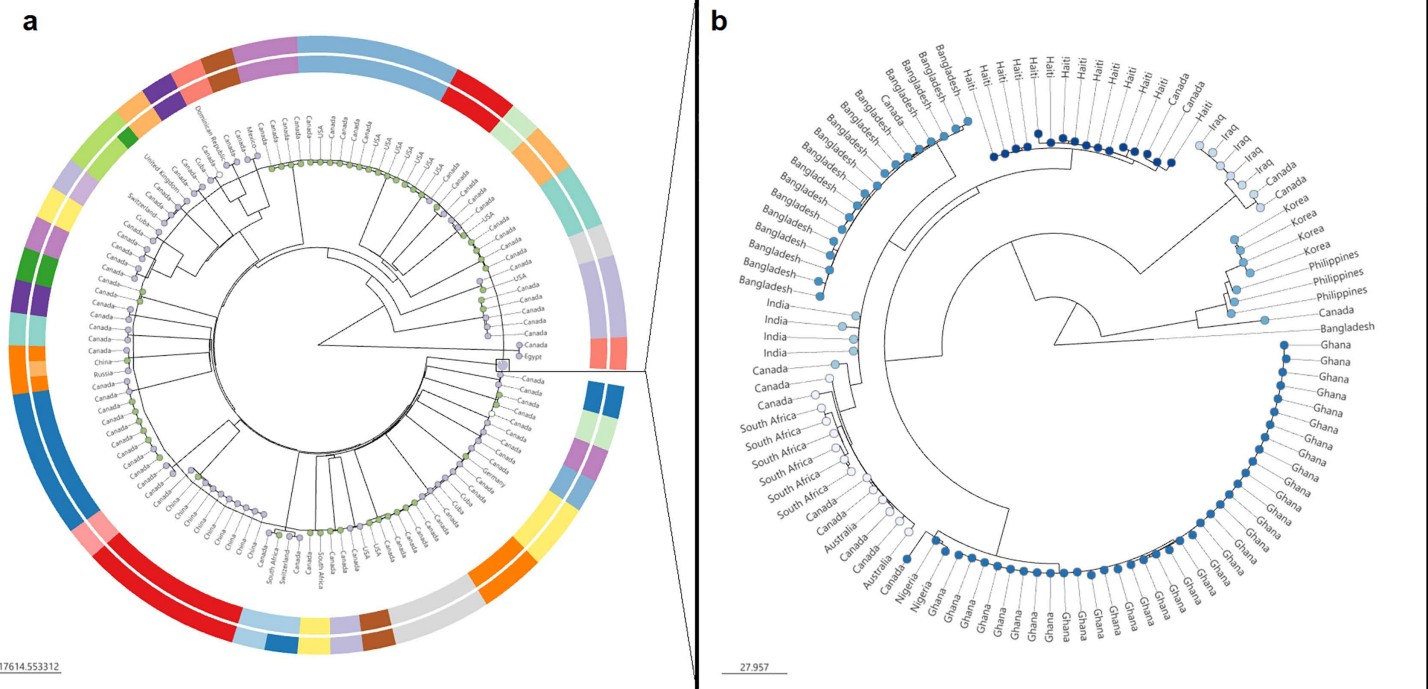

**Fig 5. Neighbour-joining phylogenetic tree of 223 global *V. cholerae* isolates.** The tree was reconstructed using the cgMLST scheme established by Liang and others [48] and plotted using Phylocanvas [121] in Vibriowatch. a. Inner ring represents clustering by sequence type based on the 7-gene MLST scheme by Octavia and colleagues [55], whereas the outer ring represents clustering based on the sublineage threshold (i.e., 133 allelic difference) designated by Liang and colleagues [48]. Tree nodes are colored by the host in which the isolate originated (green = environmental, purple = clinical). All groups inside the 7th pandemic lineage have been collapsed and are shown in the popup box (b). b Outbreak clusters (≤ 40 allelic differences) and geographical origin of *V. cholerae* serogroup O1 collected in Canada. Tree nodes are colored by outbreak clusters. An interactive version of this figure is available on Microreact (https://microreact.org/project/cLqNBRRphmNt1aoBeh9jmJ-fig-5)..

related environmental-clinical cluster to come from such different geographical locations [48]. This could be a result of an insufficient representation of NOVC and environmental sampling of *V. cholerae* worldwide. These results highlight the need for enhanced surveillance of NOVC focusing on retail food and environmental sources.

Despite the overall wide diversity of clinical isolates in Canada, there were several clinical pairs that were either genetically identical (0–3 allelic differences), very closely related (≤ 40 alleles) or closely related (≤ 133 alleles; Figs 2 and 5). For example, two individuals harbored an identical strain of NOVC collected two months apart in 2007 from the same province. There was no data on familial or household links between these individuals and to our knowledge, the cases were unrelated. Two additional pairs carried identical strains of NOVC, even when isolated from different years and regions of Canada. The genetic data provided sufficient evidence that these individuals most likely acquired NOVC from a common source; ongoing surveillance with an epidemiological investigation could have led to the identification of a common exposure source in each pair of cases.

There were several clinical cases of NOVC that were closely related to other clinical isolates collected from the Dominican Republic, China, Cuba, Egypt, Mexico, Russia, South Africa, or Switzerland (Fig 5a). Notably, one clinical isolate of NOVC from 2012 was very closely related (38 alleles) to the oldest known live *V. cholerae* isolate that is publicly available. This historic isolate, NCTC 30, was recovered in 1916 from a British soldier in Egypt and is believed to be of serogroup O2 [122]. Earlier reports questioned whether NCTC 30 was a member of the *V. cholerae* species because of its unusual phylogenetic positioning [123]. However, as more sequencing data becomes available phylogenetic analysis has proven

NCTC 30 belongs to a small distinct clade of non-toxigenic O139 and NOVC isolates from Haiti (n = 2) [124], Mexico (n = 2) [111], India (n = 3) [125] and now Canada (n = 1; This study) collected between 1991 and 2014 [122]. Since so few isolates and genomes are available from the time period of NCTC 30, continued genomic surveillance of NOVC will prove valuable for future evolutionary studies of *V. cholerae*.

**Environmental reservoirs of NOVC are genetically linked to probable extraintestinal manifestations of vibriosis**

CAD-1 was the only lineage that contained clinical isolates that originated from samples typically collected to diagnose extraintestinal manifestations of vibriosis (ear, n = 6; tissue/wound, n = 1; urine, n = 1). The isolates in this lineage are related by a median of 453 allelic differences in the core genome (range 31−833) and were collected over a long period of time (2002–2023). Three of the ear isolates were collected from the right ear of the same individual on the same day, but were found to have detectable within-host diversity (103–132 alleles). *V. cholerae* diversity within a single host could arise from strains diverging prior to entering the host, during an extended (chronic) infection period or by coinfection of multiple diverse strains [126]. On top of the eight clinical isolates, three environmental isolates originating from fish were included in CAD-1. Together, these isolates formed clade three using Fastbaps (Fig 4), and were collected from the Prairie provinces (Alberta, n = 10; Saskatchewan, n = 1). None of the clinical cases in CAD-1 that resulted in probable extraintestinal illnesses had a history of travel outside of Canada or to a province that is located near a marine or coastal ecosystem. These results point to the speculation that these NOVC isolates potentially originated from a type of inland water (freshwater, subsaline, or saline) including exposure to a river, lake, or pond. Interestingly, the majority of CAD-1 isolates were collected between September and February when the average daily temperature in Alberta and Saskatchewan ranges from 13°C to −15°C. During the fall and winter months, recreational water activities are typically at their lowest and water temperatures are not conducive to NOVC. It is possible that these individuals acquired NOVC in the Summer months and suffered chronic infection prior to seeking medical attention [12]. Indoor and outdoor swimming pools or hot tubs could also serve as a possible transmission route of NOVC extraintestinal infections [127,128]. CAD-1 was the only lineage that carried the heat-stable enterotoxin gene, *stn*, which was rarely identified among the other NOVC isolates (2%, n = 4). Both outer membrane protein encoding genes (*ompT* and *ompU)* were also identified among CAD-1 isolates. While none of the T3SS encoding genes were found in CAD-1 isolates, 82% carried either one T6SS encoding gene (*vasX* or *vgrG*). These results show that there likely is a recurring aquatic source of NOVC that is favorable to causing extraintestinal infections.

Another environmental NOVC isolate collected from a canine in 2011 showed 102 allelic differences compared to a clinical isolate collected from an ear sample in 2023; both of which were collected in eastern provinces (Prince Edward Island and New Brunswick, respectively). The canine and human isolate had identical virulence profiles including *ompT* and *chxA*, both of which have shown to favor isolates of environmental origin and extraintestinal infections. These isolates provide an example of a NOVC that are circulating locally in the aquatic environment of eastern Canada that otherwise have not been frequently detected. Physicians should be aware of NOVC as an emerging pathogen and its extraintestinal clinical presentations, especially when patients have a history of water exposure.

**Seafood-associated NOVC has linkages to the global food supply**

Isolates belonging to CAD-3 and CAD-7 lineages were isolated from raw oysters and mussels sold at Canadian retailers between 2018 and 2019. The genetic relatedness of isolates within CAD-3 and CAD-7 lineages were only 0–7 alleles, which is consistent with the outbreak isolates characterized in CAD-2 (0–2 allelic differences). Moreover, there were > 2,000 alleles between lineages. While the Canadian phylogeny did not reveal any related human isolates to these genomes (Fig 2, > 2,000 alleles), there were five environmental isolates from the United States that were closely related (5–16 alleles) to CAD-7 (Fig 5a). The five US isolates were isolated from food and food production sources (i.e., tilapia, shrimp, a utensil, and ice) in the same years as CAD-7. Isolates of CAD-3 belonged to a novel ST and were not related to any international genomes. The cholix toxin encoding gene, *chxA*, and *ompT* were found in all isolates of CAD-3 and

CAD-7 and only in 33% and 32% of the NOVC not assigned to a lineage, respectively. CAD-7 also carried *nanH* and *vasX* of the T6SS. The *chxA*, *ompT*, *nanH*, and *vasX* genes have all been detected in clinical NOVC isolated from patients in Canada.

There were several more clinical and environmental NOVC isolates from Canada, unassociated with a lineage, but otherwise were genetically related to environmental sources of NOVC in Canada and internationally (Fig 5a). NOVC isolated from Canadian retail shrimp imported from India and Thailand were closely related to NOVC isolated from shrimp in South Africa (61 alleles) and from the US (79 alleles), respectively. It is unknown whether the shrimp collected from South Africa and the US were harvested domestically or imported, but environmental isolates from the same geographical origin tend to share a common source, especially at the sublineage level (≤ 133 allelic differences) [48].

Seven Canadian NOVC clinical isolates were separately genetically linked to environmental isolates from China, Germany, South Africa, or the United States (i.e., water, unspecified aquatic products, freshwater fish production facility). The majority of these links were between Canada and the US. Notably, a clinical isolate collected in 2006 from British Columbia was highly related (6–7 alleles) to three environmental samples collected in 2018 from a freshwater tilapia production facility in Washington, United States; a state that borders directly under British Columbia and shares a similar aquatic ecosystem. Another clinical isolates collected in 2007 formed a sublineage cluster that included clinical and environmental isolates (i.e., unspecified aquatic product) from China (n = 9) and were linked to a gastroenteritis outbreak of NOVC in China in 2016 [129]. With no records of overseas travel, it is challenging to speculate on transmission; it is possible that patients may have consumed seafood imported from their respective countries or from a similar location in which *V. cholerae* was isolated from nearby aquatic ecosystems.

Seafood, including fish and shellfish (e.g., oysters, clams and shrimp) can serve as a potential transmission source of *V. cholerae* to humans and has been linked to several seafood-borne outbreaks worldwide [27,130,131]. These results highlight the ongoing risk of imported and locally harvested aquatic products sold at commercial supermarkets as potentially pathogenic sources of *V. cholerae* infection. Consumers should be warned about the potential risks associated with the handling and consumption of raw or undercooked seafood.

## Clinical and environmental NOVC possess similar antimicrobial resistance profiles in Canada

*In silico* analysis of all 242 sequences was performed to extract antimicrobial-associated genes and infer resistance profiles using the Vibriowatch tool (Fig 6). The highest resistance was detected against trimethoprim (100%), from acquisition of one of the *dfrA* genes. In the past, trimethoprim-sulfamethoxazole (SXT) was an effective therapy for cholera patients, but *V. cholerae* has since developed high levels of resistance to SXT as well as many other antibiotics, worldwide [132]. Among the 226 NOVC isolates, 28% (n = 64) harbored the *varG* gene which infers multidrug resistance to the beta-lactam class antibiotics including carbapenems, cephalosporins (i.e., broad spectrum, ceftazidime, ceftriaxone, and cefepime) and ampicillin. Overall, antimicrobial resistance gene profiles were not significantly different between clinical and environmental NOVC isolates ($p > 0.05$; Fig 6a). Similar to the virulence profiles, the predicted resistance profiles of the clinical isolates was most aligned with environmental isolates originating from food samples (Fig 6b). It was interesting that a significantly higher proportion of beta-lactam (45%, $p < 0.01$), azithromycin (9%, $p < 0.001$) and ampicillin (45%, $p < 0.01$) resistance genes were detected among NOVC originating from the environment than from clinical infections (23%, 0%, 23%, respectively). Seven percent (n = 16) of NOVC isolates carried the *sul1* and *sul2* genes conferring resistance to sulfonamides (e.g., sulfamethoxazole and sulfisoxazole). Of the NOVC isolates, inferred resistance was detected to ciprofloxacin (27%), nalidixic acid (25%), chloramphenicol (13%), streptomycin (7%), tetracycline (4%), antiseptics (4%), and azithromycin (2%). Moreover, 15% of NOVC carried at least one of the AMR variants *gyrA* (S83I) and *parC* (S85L), predicted to confer resistance to quinolones (e.g., ciprofloxacin, nalidixic acid). The World Health Organization currently recommends tetracyclines as the first-line antibiotics against *V. cholerae* infections, and the second-line alternatives are ciprofloxacin and azithromycin [133]. The percentage of isolates that exhibited intrinsic resistance against members of the

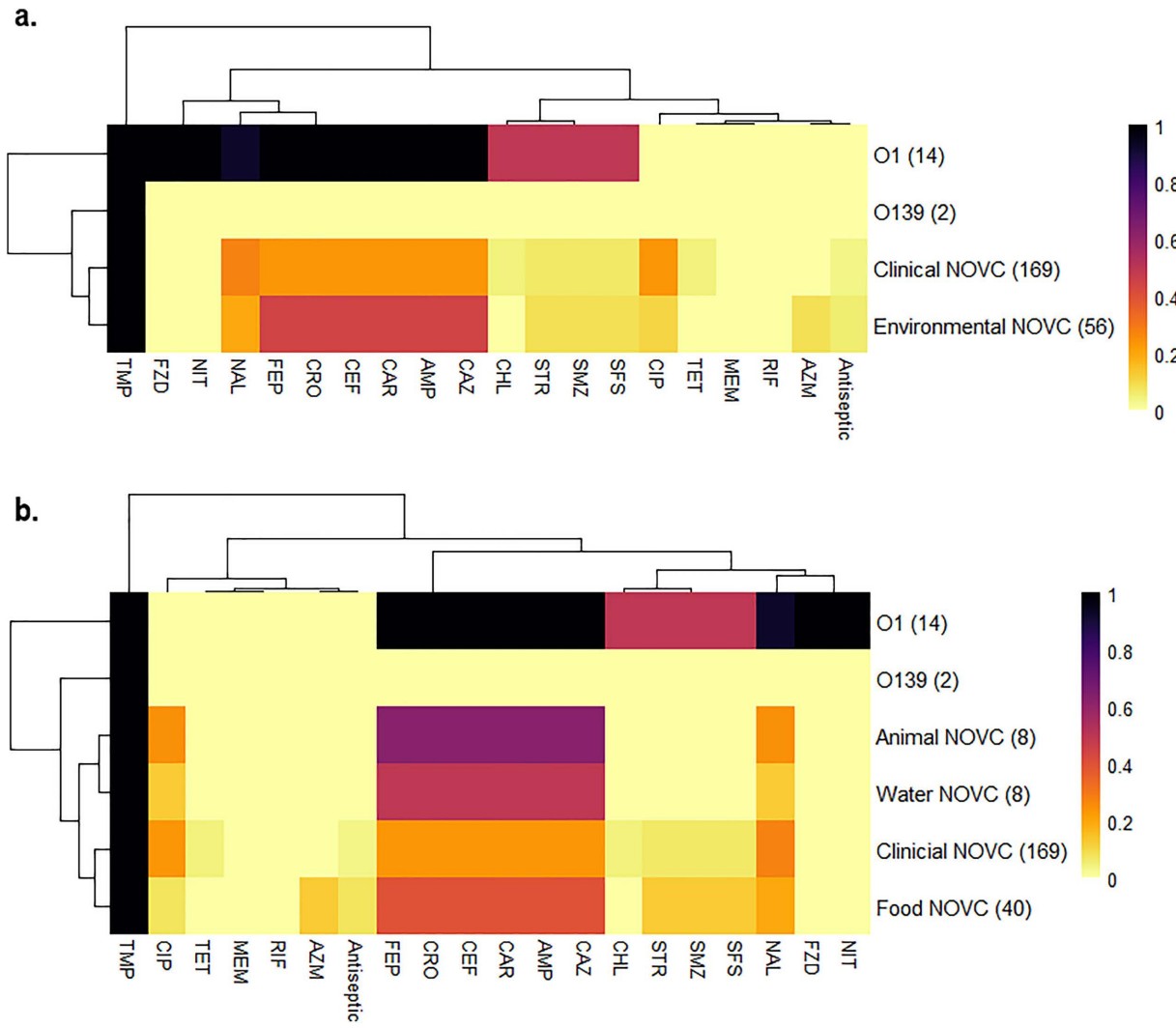

**Fig 6. Predicted antimicrobial resistance among *V. cholerae* isolated in Canada.** a. NOVC by host information. b. NOVC by detailed host information. Scale bar represents proportion of resistance present (1) or absent (0) among antibiotic classes based on resistance gene presence or absence; TMP, trimethoprim; nitrofurans (FZD, furazolidone; NIT, nitrofurantoin); TET, tetracycline; MEM, meropenem; RIF, rifampicin; AZM, azithromycin; sulfonamides (SMZ, sulfamethoxazole; SFS, sulfisoxazole); CHL, chloramphenicol; STR, streptomycin; CIP, ciprofloxacin; NAL, nalidixic acid; beta-lactams (FEP, cefepime; CRO, ceftriaxone; CEF, broad spectrum cephalosporins; CAR, carbapenems; AMP, ampicillin; CAZ, ceftazidime). One NOVC isolate did not have a recorded origin.

three recommended antibiotic classes was low; similar to what has been reported in phenotypic antimicrobial resistance studies of *Vibrio* species in Canada [134,135]. Only clinical isolates of NOVC (n = 8) were resistant to tetracyclines, seven of which also displayed intermediate or complete resistance to ciprofloxacin. Moreover, five NOVC isolates (2%) originating from imported shrimp sold at Canadian retailers carried the IncC plasmid, a known spreader of multidrug resistance [136], and displayed complete and intermediate resistance to azithromycin and ciprofloxacin, respectively, but still showed susceptibility to first-line tetracyclines. Four different plasmid amplicon types were intrinsically identified among NOVC isolates in Canada; Col(BS512) (n = 6), Col(MG828) (n = 1), Col(pHAD28) (n = 1) and IncC (n = 6). No plasmids were found in the serogroup O1 or O139 isolates.

In contrast to the resistance profiles of NOVC isolates, none of the O1 or O139 isolates had genes mediating resistance to tetracycline, ciprofloxacin, or azithromycin. Overall, serogroup O1 isolates had higher predicted resistance rates than NOVC ($p < 0.001$). Over 90% of serogroup O1 isolates harbored genes conferring resistance to nitrofurans (i.e., furazolidone, nitrofurantoin), quinolones (i.e., nalidixic acid) and beta-lactams. Half of the O1 isolates had resistance genes to chloramphenicol, streptomycin, and sulfonamides. No meropenem or rifampicin resistance genes were detected.

## Genomes of toxigenic *V. cholerae* O1 recovered from cholera cases in Canada are genetically linked to international travel

Canada is not considered endemic for cholera; previous cases were sporadic and all associated with travel. Among the 242 Canadian isolates of *V. cholerae* included in this study only two were serogroup O139 and 14 were serogroup O1; four O1 isolates were serotype Inaba, and ten were Ogawa. All of the O1 isolates were of clinical origin and associated with international travel. Travel history was only available for four O1 cases which reported travel to Haiti, Cameroon or Pakistan. Of the O139 isolates, only one was associated with international travel (i.e., Kenya), while the second isolate was acquired domestically from a wound infection. Imported cases of *V. cholerae* O139 have previously been reported in the United States [137], while Mexico suspects a domestically acquired case coming from an aquatic source [138].

Fastbaps clade three formed a monophyletic clade of strictly serogroup O1 isolates, and were all members of the 7PET phylogenetic lineage (Fig 3). While these serogroup O1 isolates were collected in Canada, they were all associated with international travel to regions where the 7PET lineage is endemic or had ongoing outbreaks. Toxigenic O139 isolates typically cluster with the 7PET lineage [83], however, the two serogroup O139 isolates in this study were non-toxigenic and dispersed among NOVC isolates in clade five.

To better understand the relationships within the 7PET lineage the 14 serogroup O1 isolates collected from unrelated cholera cases in Canada were investigated using cgMLST and placed into seven outbreak clusters (≤ 40 alleles). Each cluster genetically linked the cholera cases recovered in Canada to areas with ongoing outbreaks or endemic cholera including Pakistan (n = 6) [139,140], Iraq (n = 2) [141], India (n = 1) [142], Philippines (n = 1) [143,144], Bangladesh (n = 1) [145], West Africa (n = 1) [146,147], and Haiti (n = 2) [148,149] (Fig 5b). All 14 serogroup O1 isolates contained the CT-producing *ctxAB* genes, the main virulence factors of CTXφ such as *zot* and *ace,* and the 7PET-specific gene (VC2346). All O1 isolates also carried the *tcpA* gene, a critical receptor for *ctxAB*. Over 85% of serogroup O1 isolates contained *tagA*, *mshA*, *ompU, ompT, makA, nanH* and *vasX*, while *chxA, stn, tdh, vgrG* and all T3SS encoding genes (*vspD, vcsN2, vcsV2, vcsC2*) were not detected. All serogroup O1 isolates harbored complete VSP-1, whereas only two (14%) had the complete VSP-2. Analysis of mobile genetic elements indicated that ten (71%) of the *V. cholerae* O1 isolates harbored complete versions of both VPI-1 and VPI-2.

The two isolates belonging to serogroup O139 recovered in Canada were not genetically similar to each other or any of the other sequences from this study or within the Vibriowatch database. One O139 isolate was recovered from a clinical case that reported recent travel outside of Canada, however, there was no genetic evidence linking this isolate to a specific country or region. The second case caused by O139 did not report any travel, which is only the second time in Canada an isolate belonging to serogroup O1, or O139 has not been associated with travel. While it is unusual to find serogroup O139 outside of Asia, both isolates were non-pathogenic and did not produce CT, which typically causes outbreaks and epidemics.

## Limitations and future directions

The authors would like to acknowledge some limitations. First, this study does not provide a complete representation of NOVC in Canada due to the under-reporting of *V. cholerae* infections; the mild, self-limiting nature of vibriosis combined with its non-notifiable disease status in Canada means that the annual reports are likely an underestimation of the true number of human infections. This also explains why *V. cholerae* was not collected from all 10 Canadian provinces.

Second, the deficiency of epidemiological data for vibriosis not only for the Canadian isolates but on a global scale greatly limits our knowledge of key elements of transmission such as routes of exposure, geographical and epidemiological patterns associated with *Vibrio* infections. Future exploration of evolutionary and geographical patterns will enhance our understanding on the persistence of NOVC lineages in Canada. This study included limited environmental samples and data and thus future studies with *V. cholerae* genomes of environmental origin would allow for an integrated "One Health" approach to genomic surveillance. Moreover, it is possible that a select few NOVC isolates may have been travel-related but this information could have been missed or not collected. Thus, for the purpose of this study it is assumed that no reported travel is equivalent to no travel, and the illness was acquired domestically.

A great variety of methods and bioinformatics tools were used in this study. Lastly, the authors acknowledge that there are limitations when analysing bacterial datasets that are run on third-party web-based platforms, such as concerns with privacy, data sharing, static reference databases, misinterpretation of results and suitability for large datasets [150]. However, there are major advantages of applying online and third party platforms for WGS analysis. *In silico* typing is a fast, efficient, affordable and reproducible process. Easy to use all-encompassing and standardized workflows that are widely accessible and require little to no bioinformatics knowledge is part of the vision of PulseNet International for an ideal system to facilitate global foodborne disease surveillance using WGS [151].

## Conclusions

This study has provided a snapshot of the genomic characteristics associated with *V. cholerae* non-O1/O139, O1 and O139 isolates collected from Canada over multiple decades. The authors have demonstrated the virulence and resistance potential as well as genetic diversity among *V. cholerae* collected from clinical and environmental samples in Canada. In summary, none of the domestically acquired NOVC isolates harbored any of the traditional toxigenic genes (*ctxAB* and *tcpA*) that were found in the travel-associated serogroup O1 isolates. Several additional virulence genes were detected among NOVC including *hlyA, hapA, rtxA, toxR, nanH,* outer membrane proteins (*ompU/T*), *chxA, makA, stn, tdh, tagA, zot, ace* and secretion systems (T3SS and T6SS). NOVC and serogroup O139 isolates exhibited greater genomic diversity compared to serogroup O1 isolates. In total, 45 of 225 NOVC isolates clustered into eight novel Canadian lineages of clinical origin, environmental origin or both, indicating possible relationships between them. NOVC from seafood and freshwater environments are potential sources of vibriosis in Canada and could lead to gastroenteritis and other extraintestinal infections. Six percent of NOVC isolates were resistant to the recommended first or second-line antibiotics, which could complicate the treatment of severe infections. The emerging threat of NOVC in temperate regions, probably impacted by climate change, needs proactive monitoring and surveillance.

## Supporting information

**S1 Table. NCBI accession numbers for all study genomes.**
(XLSX)

**S2 Table. Meta-information for all Canadian isolates used in this study.**
(XLSX)

## Acknowledgments

We gratefully acknowledge the National Microbiology Laboratory Genomics Core Facilities sequencing team, laboratory technicians and managers for their assistance in this project. We are grateful to Sierra Ives from the NML and Dr. Fabiola Garcés Ayala from InDRE for excellent technical assistance of bacterial DNA and genome sequencing preparation. We thank Jillian Ryz and Matthew Wells for their guidance in bioinformatics analysis. We are also grateful for the PulseNet Canada Steering Committee member laboratories for providing the isolates used in this study, and for reviewing the

manuscript. We thank all participants of the National Institute for Communicable Diseases (NICD) GERMS-SA Laboratory Surveillance Network, South Africa for submission of clinical isolates to the NICD. The authors are also grateful to the PulseNet Latin America and the Caribbean (PNLAC) network.

## Author contributions

**Conceptualization:** Taylor Wells, Natalie Knox, Celine Nadon.

**Data curation:** Taylor Wells, Elizabeth González-Durán, Anthony M. Smith, Swapan K. Banerjee, Sandeep Tamber.

**Formal analysis:** Taylor Wells.

**Funding acquisition:** Celine Nadon.

**Investigation:** Taylor Wells.

**Methodology:** Taylor Wells, Natalie Knox.

**Project administration:** Taylor Wells, Celine Nadon.

**Resources:** Elizabeth González-Durán, Anthony M. Smith, Swapan K. Banerjee, Sandeep Tamber, Celine Nadon.

**Software:** Taylor Wells.

**Supervision:** Celine Nadon.

**Visualization:** Taylor Wells.

**Writing – original draft:** Taylor Wells.

**Writing – review & editing:** Taylor Wells, Elizabeth González-Durán, Anthony M. Smith, Swapan K. Banerjee, Sandeep Tamber, Natalie Knox, Celine Nadon.

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
