## [Decision Letter · Decision Letter 0]

12 Jul 2025

Dear Dr. Wells,

Thank you for submitting your manuscript to PLOS ONE. After careful consideration, we feel that it has merit but does not fully meet PLOS ONE’s publication criteria as it currently stands. Therefore, we invite you to submit a revised version of the manuscript that addresses the points raised during the review process.

We look forward to receiving your revised manuscript.

Kind regards,

Rahul Shubhra Mandal, Ph.D.

Academic Editor

PLOS ONE

**Journal Requirements:**

1. When submitting your revision, we need you to address these additional requirements. Please ensure that your manuscript meets PLOS ONE's style requirements, including those for file naming. The PLOS ONE style templates can be found at https://journals.plos.org/plosone/s/file?id=wjVg/PLOSOne_formatting_sample_main_body.pdf and https://journals.plos.org/plosone/s/file?id=ba62/PLOSOne_formatting_sample_title_authors_affiliations.pdf 2. Thank you for uploading your study's underlying data set. Unfortunately, the repository you have noted in your Data Availability statement does not qualify as an acceptable data repository according to PLOS's standards. At this time, please upload the minimal data set necessary to replicate your study's findings to a stable, public repository (such as figshare or Dryad) and provide us with the relevant URLs, DOIs, or accession numbers that may be used to access these data. For a list of recommended repositories and additional information on PLOS standards for data deposition, please see https://journals.plos.org/plosone/s/recommended-repositories. 3. We note that Figure 1 in your submission contain map images which may be copyrighted. All PLOS content is published under the Creative Commons Attribution License (CC BY 4.0), which means that the manuscript, images, and Supporting Information files will be freely available online, and any third party is permitted to access, download, copy, distribute, and use these materials in any way, even commercially, with proper attribution. For these reasons, we cannot publish previously copyrighted maps or satellite images created using proprietary data, such as Google software (Google Maps, Street View, and Earth). For more information, see our copyright guidelines: http://journals.plos.org/plosone/s/licenses-and-copyright. We require you to either present written permission from the copyright holder to publish these figures specifically under the CC BY 4.0 license, or remove the figures from your submission: a. You may seek permission from the original copyright holder of Figure 1 to publish the content specifically under the CC BY 4.0 license.   We recommend that you contact the original copyright holder with the Content Permission Form (http://journals.plos.org/plosone/s/file?id=7c09/content-permission-form.pdf) and the following text:“I request permission for the open-access journal PLOS ONE to publish XXX under the Creative Commons Attribution License (CCAL) CC BY 4.0 (http://creativecommons.org/licenses/by/4.0/). Please be aware that this license allows unrestricted use and distribution, even commercially, by third parties. Please reply and provide explicit written permission to publish XXX under a CC BY license and complete the attached form.” Please upload the completed Content Permission Form or other proof of granted permissions as an "Other" file with your submission. In the figure caption of the copyrighted figure, please include the following text: “Reprinted from [ref] under a CC BY license, with permission from [name of publisher], original copyright [original copyright year].” b. If you are unable to obtain permission from the original copyright holder to publish these figures under the CC BY 4.0 license or if the copyright holder’s requirements are incompatible with the CC BY 4.0 license, please either i) remove the figure or ii) supply a replacement figure that complies with the CC BY 4.0 license. Please check copyright information on all replacement figures and update the figure caption with source information. If applicable, please specify in the figure caption text when a figure is similar but not identical to the original image and is therefore for illustrative purposes only.The following resources for replacing copyrighted map figures may be helpful: USGS National Map Viewer (public domain): http://viewer.nationalmap.gov/viewer/The Gateway to Astronaut Photography of Earth (public domain): http://eol.jsc.nasa.gov/sseop/clickmap/Maps at the CIA (public domain): https://www.cia.gov/library/publications/the-world-factbook/index.html and https://www.cia.gov/library/publications/cia-maps-publications/index.htmlNASA Earth Observatory (public domain): http://earthobservatory.nasa.gov/Landsat:
http://landsat.visibleearth.nasa.gov/USGS EROS (Earth Resources Observatory and Science (EROS) Center) (public domain): http://eros.usgs.gov/#Natural Earth (public domain): http://www.naturalearthdata.com/

Reviewers' comments:

Reviewer's Responses to Questions

**Comments to the Author**

1. Is the manuscript technically sound, and do the data support the conclusions?

Reviewer #1: No

Reviewer #2: Yes

2. Has the statistical analysis been performed appropriately and rigorously?

Reviewer #1: N/A

Reviewer #2: Yes

3. Have the authors made all data underlying the findings in their manuscript fully available?

Reviewer #1: No

Reviewer #2: Yes

4. Is the manuscript presented in an intelligible fashion and written in standard English?

Reviewer #1: Yes

Reviewer #2: Yes

**Reviewer #1:**  Dear authors, I went through your manuscript and found some changes that must be done. Here are my suggestions-

1. The definition of a new CAD lineage based solely on a cluster of ≥3 isolates lacks methodological rigor. A more quantitative threshold (e.g., allelic divergence) should be applied alongside phylogenetic clustering to robustly delineate novel lineages.

2. While 242 Canadian isolates were analyzed, the manuscript lacks a clear rationale for focusing on Canada in a globally relevant context. Adding a comparative or translational public health perspective (e.g., impact of NOVC on coastal communities or seafood industries) would improve relevance.

3. The study relies heavily on cgSNP analysis but does not clarify the rationale for SNP thresholds used to define lineages, outbreaks, and sublineages. The cgMLST cutoffs (e.g., 3, 40, 133, 1000 alleles) need to be referenced and justified within the context of V. cholerae evolution.

4. Environmental data (e.g., temperature, salinity, sample site type) are not quantitatively integrated. A generalized linear model or spatiotemporal regression could better reveal correlations between environmental factors and lineage emergence.

5. Given the known plasticity of V. cholerae genomes, especially NOVC strains, the potential role of HGT in virulence gene dissemination (e.g., tdh, T3SS, VSP islands) is underexplored and should be evaluated using tools like ICEfinder or PlasmidFinder.

6. While the authors tabulate virulence genes, they fail to conduct any comparative pan-genome or synteny analyses. A deeper annotation of pathogenicity islands (e.g., VPI-1, VPI-2) using genome browsers would provide better resolution.

7. All conclusions about virulence are based on in silico predictions. There is no transcriptomic or phenotypic validation (e.g., hemolysin assays, T3SS effector secretion), which weakens claims about clinical relevance.

8. The AMR gene analysis is described in passing with no detailed breakdown of classes (e.g., β-lactamase variants, efflux pumps). Also, no mention is made of mobile genetic elements carrying AMR genes, which is critical in surveillance.

9. The manuscript leans heavily on third-party platforms without disclosing the limitations of these tools (e.g., static databases, annotation errors). Results should be cross-validated using independent BLAST-based methods or custom HMM profiles.

10. The manuscript misses an opportunity to discuss how NOVC evolution compares with pandemic O1/O139 strains (e.g., clonal expansion vs. recombination-driven diversification). Phylogenetic trees should include time-calibration or molecular clock estimates.

11. Student’s t-test is applied to categorical variables (gene presence/absence), which violates assumptions. A more appropriate test would be Fisher’s exact or chi-square test with FDR correction.

12. There is no accessible summary table (as a figure or supplemental file) that links isolates to year, location, lineage, serogroup, and source. This omission hinders reproducibility and reader comprehension.

13. The classification of isolates as "clinical" or "environmental" is vague and lacks metadata evidence (e.g., case records, exposure history). Misclassification could bias virulence profile comparisons.

14. Recombination filtering using PhiPack is briefly mentioned, but no recombination rate estimates or visualizations (e.g., Gubbins plots) are shown. The extent of recombination should be quantified.

15. The core genome phylogenetic trees are not sufficiently annotated (e.g., bootstrap values, gene overlays). Inclusion of interactive figures (e.g., Microreact or iTOL links) is suggested but not demonstrated.

16. While climate change is a plausible driver of Vibrio emergence, the manuscript repeatedly makes unquantified claims without supporting climate models or water temperature data from Canada.

17. Genes like tdh, ctxAB, toxR are inconsistently referred to across figures and text. A consistent gene naming system should be applied (e.g., italicized for genes, capitalized for proteins).

18. The manuscript misses the chance to link Canadian findings to broader One Health or global AMR surveillance initiatives (e.g., GISAID-equivalent for Vibrio, or WHO GLASS).

19. Custom pipelines like “mikrokondo” are not well-described or linked to repositories. Readers must have access to full GitHub/DOI-archived scripts and parameters for reproducibility.

20. Although MLST types are mentioned, the manuscript doesn’t explore the evolutionary or geographic patterns of sequence types across the Canadian or global dataset using STRUCTURE or goeBURST.

**Reviewer #2:**  Greetings

Great work.

But there is one figure about the phylogenetic tree is not clear.

Kindly try to put more clear figure.

Kindly write the scientific names of bacteria by italic line style in whole manuscript.

Kind regards.

**Do you want your identity to be public for this peer review?** For information about this choice, including consent withdrawal, please see our Privacy Policy

Reviewer #1: **Yes:** Shaban Ahmad

Reviewer #2: No

---

## [Author Response · Author response to Decision Letter 1]

18 Oct 2025

Please see rebuttal letter that responds to each point raised by the academic editor and reviewer(s).

---

## [Decision Letter · Decision Letter 1]

12 Dec 2025

Dear Dr. Wells,

Thank you for submitting your manuscript to PLOS ONE. After careful consideration, we feel that it has merit but does not fully meet PLOS ONE’s publication criteria as it currently stands. Therefore, we invite you to submit a revised version of the manuscript that addresses the points raised during the review process.

We look forward to receiving your revised manuscript.

Kind regards,

Rahul Shubhra Mandal, Ph.D.

Academic Editor

PLOS One

Journal Requirements:

Reviewers' comments:

Reviewer's Responses to Questions

**Comments to the Author**

Reviewer #1: All comments have been addressed

Reviewer #3: (No Response)

2. Is the manuscript technically sound, and do the data support the conclusions?

Reviewer #1: Yes

Reviewer #3: Yes

3. Has the statistical analysis been performed appropriately and rigorously?

Reviewer #1: Yes

Reviewer #3: Yes

4. Have the authors made all data underlying the findings in their manuscript fully available?

Reviewer #1: Yes

Reviewer #3: Yes

5. Is the manuscript presented in an intelligible fashion and written in standard English?

Reviewer #1: Yes

Reviewer #3: Yes

Reviewer #1: Dear authors,

I read the revised manuscript and found that your team has resolved everything and therefore I mark it for acceptance. Congratulations.

Reviewer #3: 1. The detection of the tdh (thermostable direct hemolysin) gene in the CAD-2 lineage—which was associated with the herring egg outbreak—is a significant finding. However, the manuscript implies this gene is linked to pathogenicity similar to V. parahaemolyticus. The authors should provide a more granular analysis of this gene sequence. Specifically, what is the nucleotide or amino acid identity of this tdh allele compared to the canonical pathogenic tdh found in V. parahaemolyticus? Confirming whether this is a functional homolog or a distinct variant is crucial for substantiating the virulence potential discussion in lines 538-547.

2. In the antimicrobial resistance section (Line 725), the authors state: 'All isolates harbored at least one of the dfrA genes conferring resistance to trimethoprim.' A 100% prevalence of acquired resistance genes across diverse lineages spanning 25 years is statistically unusual for non-O1/non-O139 isolates. The authors should clarify if the specific dfrA variants detected are known intrinsic chromosomal genes in V. cholerae or legitimate acquired resistance markers. If they are intrinsic, classifying them as markers of resistance in the context of surveillance might be misleading without qualification.

3. The authors perform statistical comparisons between 'clinical' and 'environmental' isolates (Fig 3 and text). However, the 'environmental' category aggregates 'food' (seafood/retail), 'water', and 'animal'. As shown in Figure 3b, food isolates appear to share more virulence features with clinical isolates than water isolates do. Grouping retail seafood (which acts as a vector) with environmental water samples might dilute statistically significant differences regarding ecological adaptation vs. infectivity. I suggest a brief sub-analysis or discussion point distinguishing 'natural environment' (water/sediment) from 'food/retail' to refine the conclusion that clinical and environmental isolates differ.

4. The authors identify eight 'new' lineages (CAD1-8) and state they 'represent populations of NOVC local to Canada' (Line 519). While the phylogenetic clustering is sound based on the cited cgMLST scheme, the claim of locality should be tempered. Given the vast under-sampling of NOVC genomes globally (specifically from environmental sources in other nations), it is highly probable these lineages circulate elsewhere but simply haven't been sequenced. I recommend modifying the text to acknowledge that these are 'lineages currently unique to the Canadian dataset' rather than definitively local or endemic to Canada.

5. While the inclusion of Microreact links is excellent for transparency, the static versions of Figure 2, Figure 4 and Figure 5 included in the manuscript are very dense and difficult to read. In Figure 2, the heatmap columns are indistinguishable without zooming in significantly, and the tip labels are illegible. I recommend providing a simplified or 'collapsed' version of the tree for the main text that highlights the CAD lineages and major clades, while moving the fully annotated, dense trees to the Supplementary Materials.

**Do you want your identity to be public for this peer review?** For information about this choice, including consent withdrawal, please see our Privacy Policy

Reviewer #1: No

Reviewer #3: **Yes:** Dr. Debaki Ranjan Howlader

---

## [Author Response · Author response to Decision Letter 2]

24 Jan 2026

Dear Editor and Reviewers,

Thank you for the thorough review and constructive feedback you provided on my manuscript submitted to PLOS ONE. Your insightful comments have been invaluable in improving the quality and clarity of the manuscript. I have carefully considered each of your suggestions. Author's point-by-point responses to the journal requirements and reviewer’s feedback can be found in a separate document attached to this submission. We hope our revised version meets your expectations.

Thank you once again for your time, effort, and valuable contributions to the advancement of this work.

Best regards,

Taylor Wells

Corresponding Author

taylor.m.wells@phac-aspc.gc.ca

---

## [Decision Letter · Decision Letter 2]

4 Feb 2026

Population structure and phylogenetic analysis of Vibrio cholerae non-O1/O139 by whole genome sequencing

PONE-D-25-23191R2

Dear Dr. Wells,

We’re pleased to inform you that your manuscript has been judged scientifically suitable for publication and will be formally accepted for publication once it meets all outstanding technical requirements.

Kind regards,

Rahul Shubhra Mandal, Ph.D.

Academic Editor

PLOS One

Additional Editor Comments (optional):

Reviewers' comments:

Reviewer's Responses to Questions

**Comments to the Author**

Reviewer #3: All comments have been addressed

2. Is the manuscript technically sound, and do the data support the conclusions?

Reviewer #3: Yes

3. Has the statistical analysis been performed appropriately and rigorously?

Reviewer #3: Yes

4. Have the authors made all data underlying the findings in their manuscript fully available?

Reviewer #3: Yes

5. Is the manuscript presented in an intelligible fashion and written in standard English?

Reviewer #3: Yes

Reviewer #3: (No Response)

**Do you want your identity to be public for this peer review?** For information about this choice, including consent withdrawal, please see our Privacy Policy

Reviewer #3: **Yes:** Debaki Ranjan Howlader

---

## [Editor Report · Acceptance letter]

PONE-D-25-23191R2

PLOS One

Dear Dr. Wells,

I'm pleased to inform you that your manuscript has been deemed suitable for publication in PLOS One. Congratulations! Your manuscript is now being handed over to our production team.

Kind regards,

on behalf of

Dr. Rahul Shubhra Mandal

Academic Editor

PLOS One